# Practitioner Rehabilitation following Professional Misconduct: A Common Practice Now in Need of a Theory?

Lois J. Surgenor [1,*], Kate Diesfeld [2] and Marta Rychert [3]

1 Department of Psychological Medicine, University of Otago, P.O. Box 4345, Christchurch 8140, New Zealand
2 School of Public Health and Interpersonal Studies, 90 Akoranga Drive, Auckland 0627, New Zealand
3 SHORE and Whariki Research Centre, Massey University, 7/90 Symonds Street, Auckland 1010, New Zealand
* Correspondence: lois.surgenor@otago.ac.nz

**Abstract:** Theories of rehabilitation have long been articulated in health and criminal justice contexts, driving rehabilitation practices in each area. In this article, several prominent theories are described to illustrate how their core assumptions aim to facilitate recovery and reduce relapse or reoffending. Professional disciplinary bodies are also often compelled by law or regulation to attend to practitioners' rehabilitation after professional misconduct, with similar aims to restore the practitioner to safe practice. Yet, no rehabilitation theory has been articulated in this context despite professional rehabilitation being distinct from other settings. We propose that the current absence of a coherent theory is problematic, leaving professional disciplinary bodies to 'borrow' assumptions from elsewhere. Since rehabilitation penalties are frequently made by professional disciplinary bodies, we review several theories from health and justice contexts and highlight elements that may be useful in developing professional misconduct rehabilitation theory. This includes proposing methodological approaches for empirical research to progress this.

**Keywords:** professional; misconduct; rehabilitation; penalties; disciplinary; tribunals

## 1. Introduction

The term 'rehabilitation' has multiple meanings but a common understanding is that it involves restoring someone to a previous, relatively better state. The notion is often applied to: restoration of health (e.g., physical rehabilitation after an accident or illness); behaviour (e.g., rehabilitation of offenders in criminal justice systems); employment; and professional roles. In health contexts, the World Health Organization's widely used definition is that rehabilitation is "a set of interventions designed to optimise functioning and reduce disability in individuals with health conditions in interaction with their environment" (https://www.who.int/news-room/fact-sheets/detail/rehabilitation, accessed on 3 June 2023). This is intended to cover broad ranging health impairments.

Strong arguments for rehabilitation also apply in the professional disciplinary context and such rehabilitation features in disciplinary decisions in many professions across many countries. For example, in New Zealand, the prospect of rehabilitation is one of the penalty principles that the Health Practitioners Disciplinary Tribunal (HPDT) "cannot ignore" (A v PCC (5 September 2008, Keane J, CIV-2008-404-2927, at 27). Likewise for lawyers, failure to consider rehabilitative conditions has been described as leaving the disciplinary body's job "largely undone" (paragraph 17, https://www.justice.govt.nz/assets/2021-NZLCDT-27-Auckland-Standards-Committee-4-v-OBoyle.pdf accessed on 6 June 2023).

Similar obligations are evident in other international jurisdictions. For example, in the United Kingdom, whether the misconduct can or cannot be addressed are factors the Nursing and Midwifery Council uses in determining sanctions which include retraining or other assessments (https://www.nmc.org.uk/ftp-library/understanding-fitness-to-practise/insight-and-strengthened-practice/can-the-concern-be-addressed, accessed on 18 May 2023). In Australia, rehabilitative steps to guard against recurring

professional misconduct are factors taken into account by the Queensland Civil and Administrative Tribunal. For example, following a finding of professional misconduct, the Tribunal considered a lawyer's rehabilitative steps and ordered that he continue consulting a psychologist for as long as therapeutically necessary for him to function appropriately in legal practice (Legal Services Commissioner v Cruise [2019] QCAT 182; https://www.lsc.qld.gov.au/__data/assets/pdf_file/0015/620232/cruiseocr157-17.pdf, accessed on 18 May 2023).

In this paper, we first describe the widely argued benefits of rehabilitation, followed by examples of selected rehabilitation theories in health and criminal settings. With each example, we discuss how this might apply in professional misconduct rehabilitation. Thereafter, the benefits of developing a professional misconduct rehabilitation theory are argued, including what may be expected of minimal requirements of such theory. The paper concludes with consideration of the types of research which can inform such theory development.

## 2. Benefits of Rehabilitation

Rehabilitation is considered beneficial for individuals and society, and researchers have argued that access to rehabilitation is a basic human right (Siegert et al. 2010). What qualifies as a good rehabilitation outcome is theory-driven, context-dependent, and debatable (McPherson et al. 2010), as is how best to measure these outcomes (Kayes and McPherson 2010). However, several benefits recurred across definitions in the literature.

It is commonly argued that applying rehabilitation theory impacts the speed, degree, or type of 'recovery'. This assumption is especially evident in health and injury literature (McPherson et al. 2015) as well as drug and alcohol rehabilitation (Best and Lubman 2012). Empirical evidence in support of the impacts on recovery ranges from 'strong' to 'medium' (e.g., McCarthy et al. 2015) to 'weak' effects (e.g., see Peters et al. 2021), depending on the population and context studied. One key factor that explains the variability of rehabilitation's effects on health recovery is that many studies have tested hypotheses without reference to any specific rehabilitation theory (Dunn and Elliott 2008). Moreover, the broad reach of rehabilitation research regarding health covers many diverse settings and populations. Further, not all conditions are equally amenable to rehabilitation.

The effect of rehabilitation on the speed, degree, and quality of 'recovery' is also made in criminal contexts. However, there are variations in what constitutes 'recovered' across the diverse studies. For example, recovery may focus on reduction in risk factors (e.g., substance dependence) that contribute to criminal acts. Some studies use rates of relapse as a proxy for rehabilitation, through measures of recidivism, arrests, and parole violations. For example, to prevent recidivism, the criminal justice system may provide behaviour change programmes to reduce sexual offending and violence or self-support skills programmes for vocational and educational purposes. As with health settings, a large number of studies have explored if and how rehabilitation aids recovery after offending or reduces rates of relapse (e.g., Perry et al. 2019).

Aside from personal benefits to the rehabilitated, rehabilitation in both health and criminal settings is also linked to wider social benefits. These include financial savings, reduced length of hospital stays (e.g., Thomas et al. 2019), and later employability on re-entering society (e.g., Varghese et al. 2010). There are other reasons for ordering rehabilitation penalties. They include preserving the investment in practitioners' training (e.g., Roberts v PCC of the Nursing Council of New Zealand, https://www.hpdt.org.nz/portals/0/613HP13265P.pdf, accessed on 30 January 2023), preserving the practitioner's livelihood (e.g., 2020-NZLCDT-42-Nelson-Lawyers-Standards-Committee-v-Stevenson.pdf, accessed on 18 May 2023) and signalling to the profession that factors involved in misconduct can be addressed (Stemwedel 2014) and are not career-ending.

## 3. Rehabilitation Theory Examples

Attempts to develop a single theory applying to all rehabilitation contexts have been described as "at best ambitious and at worst futile" (McPherson et al. 2015, p. 9) Thus, in health, disability, and criminal contexts there are a number of commonly cited rehabilitation theories with differing ideological underpinnings, values, and practices. The following section describes commonly used rehabilitation theories and sets the scene for crafting a rehabilitation theory in the professional misconduct context. The aim is not to argue for the superiority of one existing theory. Nor is the aim to establish the necessary conditions leading to rehabilitation. Rather, it argues that there is a need for theory designed for the specific context of regulated professional practitioners who are ordered to undertake rehabilitation after misconduct.

### 3.1. Theories of Rehabilitation in Health Settings

3.1.1. Person-Centred Rehabilitation

Mainly applied in health contexts, a person-centred model of rehabilitation (PCR) has been articulated (Health Innovation Network–South London 2016) with purported broad-ranging outcomes. PCR emphasises particular attributes and roles of those delivering and receiving rehabilitation. For example, rehabilitation is respectful of, and tailored to, the person through the focus on meanings, hope, and strengths. Moreover, rehabilitation is inclusive of significant others and that it is essential for practices of rehabilitation to occur in a welcoming and secure environment. In short, PCR places emphasis on how the person and rehabilitation provider are prioritized; its central philosophy is that rehabilitation is crafted with the person, rather than prescribed and delivered to the person (Jesus et al. 2016). The recipient of the rehabilitation also plays an important part in evaluation and quality improvement, meaning that surveys of the experience contribute to broader organizational improvements ('macrosystems', Jesus et al. 2022).

PCR approaches are argued as central to high quality rehabilitation (Jesus et al. 2022; McPherson et al. 2015). However, some PCR proponents also acknowledge that in some settings its use may be limited, especially if the person receiving rehabilitation has insufficient insight or mental capacity to articulate appropriate needs or goals (Fernandes et al. 2022). Recent research (Jesus et al. 2022) addressed concerns about the need for greater theoretical and conceptual clarity of PCR (Scholl et al. 2014). Moreover, there was a need to acknowledge the resource implications for providers, treatment teams, and organisations. For example, "co-constructed" rehabilitation decision-making may have fiscal impacts.

If PCR was applied in the disciplinary context, it can begin by the disciplinary tribunal working collaboratively to co-design the rehabilitation orders with the practitioner (including potentially with their significant others), taking into account the whole person and not just their misconduct activity. The practitioner's own world views and preferences would not be presumed, though once understood those views/preferences would hold at least equal weight as the "expert", including the right not to agree to certain rehabilitation goals or activities. In theory, for example, if a practitioner did not see their opioid addiction as problematic and thus did not value accessing addiction treatment to reduce the risk self-prescribing opioids (the disciplinary issue), this would not be included in the rehabilitation plan. In short, the decision to not agree to that goal and refuse interventions would be respected even if it was highly valued by the tribunal, profession, and society. In implementing the co-constructed rehabilitation plan, interactions between the practitioner and the person delivering the rehabilitation (for example, an appointed supervisor) would have a compassionate and empathic stance including, where appropriate, time for reciprocal interaction of shared experiences. Outcomes of the rehabilitation would be reported but the practitioner's experience of the rehabilitation would be elicited and contribute to this.

It is difficult to know what components of the PCR model can be valued, or work, in the context of professional misconduct rehabilitation. Currently, some components may already be present, such as fostering a supportive, collaborative relationship between rehabilitation providers (e.g., supervisors and educators) and the disciplined practitioner.

However, other PCR principles and practices may be novel or controversial, especially the notion of disciplined practitioners co-constructing their rehabilitation penalty with the disciplinary tribunal. Moreover, this strategy may be objectionable to those who are harmed by the professional misconduct.

3.1.2. Common Sense Model (CSM)

This theory (Leventhal et al. 2003) is widely used in physical health, and increasingly mental health, contexts (Cannon et al. 2022). It arises from the psychological study of how people respond to health threats or conditions and strongly draws on coping theory (Carver et al. 1992).

The core model proposes that rehabilitation is influenced by several factors. First, rehabilitation is impacted by beliefs regarding health and illness. These include: beliefs about the consequences of having a health condition; timelines needed for rehabilitation; the degree of perceived control over the health condition; and the degree to which the health condition 'makes sense' to them as the patient.

Second, rehabilitation may be affected by the person's explanations or beliefs about the cause of their health condition. For example, rehabilitation can be impacted by the degree to which a person believes their situation or condition has been caused by accident or chance, risk factors beyond their control, or even psychological attributions. By way of illustration, having stronger expectations of experiencing long-lasting symptoms is associated with increased odds of a poor recovery from mild traumatic brain injury after six months (Snell et al. 2013). Across a range of health conditions, the CSM model has been reported to predict health outcomes, including both physical and emotional health outcomes (Dempster et al. 2015; Hagger et al. 2017).

Applied in the professional misconduct rehabilitation context, CSM practice would begin by a clear assessment of what the practitioner believes was associated with, or caused, their misconduct. The rehabilitation then would be designed to address practitioners' gaps in knowledge or to correct inaccurate beliefs (e.g., 'lay explanations'). Moreover, it would address disproportional cognitive or emotional responses to the misconduct (e.g., catastrophically concluding that their career is over; steadfast denial that the misconduct was unethical). Then the rehabilitation activities (many of which are didactic) would be tailored to encourage more accurate timelines and understanding of the causes and consequences of their misconduct to foster future preventative behaviour. Thus, attending to the practitioner's overly catastrophic reactions or cognitive blind spots (e.g., rationalisations and denials) is just as important as helping the practitioner understand the highly probable cause and effect factors. This is because both these factors influence the selection of future coping strategies and behaviour.

Conceivably, some of these concepts (e.g., causal and consequence understandings; levels of perceived/accepted personal control) are already weighed by disciplinary tribunals and rehabilitation providers. For example, levels of insight about factors leading to the misconduct (suggesting levels of understanding of causal factors) is often considered in determining the nature of rehabilitative penalties (e.g., see https://www.hpdt.org.nz/portals/0/45Nur0519P.pdf, accessed on 3 May 2023). Likewise, rehabilitation may be more difficult to achieve for practitioners who believe they will never recover (CSM timeline) from the shame of their misconduct and that their damaged professional reputation is irreparable. Other core concepts in CSM (e.g., day-to-day and symptom consequences) would need to be modified, or may be less adaptable, in any professional rehabilitation theory on the basis that CSM theory is usually applied in health settings with people living with health conditions.

*3.2. Theories of Rehabilitation in Justice Contexts*

In the criminal context, rehabilitation goals and priorities are often determined by others or society generally (e.g., reduced risk of offending). In some respects, the notion of

rehabilitation in criminal settings has a distinctive orientation. Four different theories and models are discussed, while we acknowledge that others may have relevance.

### 3.2.1. Risk-Need-Responsivity (RNR)

This theory focuses on reducing criminal recidivism through a set of rehabilitation actions and principles. It is based upon a staged, structured risk and needs analysis (Bonta and Andrews 2007). Stage 1 includes assessment for risk and causal factors for the offending. Factors may include personality patterns, cognitive and emotional factors, and substance abuse. Stage 2 then identifies targets (needs) which are amenable to change. Stage 3 (responsivity) refers to tailoring (in style or mode of delivery) of those interventions to address the needs, based on consideration of the person's circumstances, including demographic or psychological ones. 'Principles' include matching intervention intensity with risk levels (Andrews and Dowden 2007). Some refer to RNR as "crime-prevention jurisprudence" (Andrews and Dowden 2007, p. 439).

The RNR model has become a leading model of offender rehabilitation in the world, and has been adopted in numerous countries (e.g., in New Zealand see https://www.corrections.govt.nz/resources/research/risk-assessment-of-recidivism-of-violent-sexual-female-offenders/risk-need-and-responsivity, accessed on 7 February 2023). Modifications to the theory have occurred (e.g., RNR-I; Looman and Abracen 2013). Research support for the core theory has been both critical (e.g., Ward et al. 2012, plus see Polaschek (2012) for an overview) and supportive (e.g., Vose et al. 2020). Reduction in recidivism attributable to the RNR approach has been reported (Dowden and Andrews 2000) though factors undermining this have also been identified (Bloom and Bradshaw 2021). Inconsistent practices in applying the necessary steps and principles have been reported as a possible reason for mixed findings (Brogan et al. 2015).

In applying RNR to professional misconduct rehabilitation, firstly this would begin with a formal assessment of the risk of repeating professional misconduct and thus posing a level of risk to the public. This would be performed through interviews and/or the use of evidenced-based measures to quantify static (e.g., previous professional misconduct in comparable circumstances) and dynamic factors (e.g., potentially modifiable issues such as substance abuse, practice circumstances, and professional isolation). Based on the level of classified risk, those with high risk may have intensive rehabilitation conditions such as one-one-one rehabilitation over a prolonged period of time. The detail and target of this would then be tailored to matters such as the practitioner's motivation, learning style, and personality. Those with low risk may not have any penalties stipulating rehabilitative conditions.

There is no explicit reference to RNR in the professional misconduct arena, though "risk" and "responsiveness" considerations (in ordering specific rehabilitation conditions) are evident in some decisions in New Zealand at least. For example, in Med10/149P the HPDT ordered a medical practitioner to comply with specific rehabilitation steps deemed necessary by a psychiatrist and/or clinical psychologist following professional discipline arising from criminal convictions for possessing objectionable material. This implies that the detail of rehabilitation is determined following the assessment of risk factors ('risk'), with consideration of what factors are amenable to change. In this way, disciplinary tribunals sometimes accept that other experts are best positioned to determine risk and the precise components of rehabilitation.

### 3.2.2. Restorative Justice

Restorative justice (RJ) has been defined as a process, a theory, and a social movement with a variety of historical roots (see Hass-Wisecup and Saxon 2018 for an historical overview). It involves a rehabilitative approach to justice "whereby all the parties with a stake in a particular offence come together to resolve collectively how to deal with the aftermath of that offence and its implications for the future" (Marshall 1999, p. 5). Based on this broad definition, there are differing views as to the key processes and concepts of RJ

theory (McCold and Wachtel 2002; Menkel-Meadow 2007). However, there is agreement that RJ is collaborative, future-orientated, and reintegrative, with victims having a say in how to repair the harm caused by the offending.

RJ aims to reintegrate victims and offenders better than traditional criminal justice systems (McCold and Wachtel 2002). RJ theory proposes that this occurs though processes which allow victims to regain some sense of control, while simultaneously offenders may build a connection with the victim (and community values) by admitting responsibility and experiencing 'reintegrative shame' (Braithwaite 1989). The role of interpersonal forgiveness remains peripheral because this is considered an emotional response that cannot be forced on victims (Suzuki and Jenkins 2022). Any resulting acceptance and forgiveness may help the victim psychologically recover (Griffin et al. 2015) which is the focus of RJ. Incidentally, it may help the practitioner to return to practice. However, this may not be the same as rehabilitating practice.

RJ has been applied across a range of criminal and organisational contexts, though there are concerns that certain offenders and crimes would be unsuitable for this approach (e.g., rape, Robinson 2003). On the other hand, RJ has been argued as more culturally appropriate than traditional justice responses across a range of communities due to its indigenous roots (Hadley 2001).

In the case of professional misconduct rehabilitation, misconduct would be determined by a disciplinary body, but can accompanied by a gathering between those who had been harmed or affected by that misconduct and the practitioner. The meeting would empower the harmed parties to have a say in the restitution of the harm done, though only if they consent. This restitution might include a 'reintegration plan' overseen by an RJ facilitator and can also include institutional or service structures and policies to reduce the risks of future harm. Crucially, the practitioner would have to take responsibility for the misconduct and engage in activity to repair the harm (Pavlacic et al. 2022).

There is already some discussion of the use of RJ in the professional misconduct rehabilitation arena. Examples include application of RJ to address sexual misconduct in education institution settings (Koss et al. 2014) or mistreatment of tertiary education students by faculty academic staff (Acosta and Karp 2018), and settling medical malpractice cases (Syauf et al. 2020). However, some commentators are concerned that transparency and accountability may be lost if matters are settled "behind closed doors" (Haller 2012, p. 332).

Further, RJ's focus on repairing relationships and the practitioner taking responsibility may limit its application in professional disciplinary contexts as not all cases of professional misconduct involve harmed victims (e.g., a self-prescribing pharmacist). Moreover, a sizable number of practitioners deny the charges (Surgenor et al. 2020). The option of conciliation between the practitioner and complainant is already available in some jurisdictions, but this appears to be rarely used and never used when the complaints are serious (Grenyer and Lewis 2012; Miller 2017; Surgenor and Diesfeld 2018). It is acknowledged that conciliation is not the same as RJ, but the above suggests that meetings between the practitioner and the aggrieved person are currently extremely rare in regulatory settings.

### 3.2.3. Good Lives Model

The Good Lives Model (GLM) (Ward 2002) was initially developed for rehabilitation of people convicted of sexual offences, but has been extended to a range of correctional and offending settings (Mallion et al. 2020) including those convicted of domestic violence (Langlands et al. 2009) and indigenous individuals in criminal justice systems (Strauss-Hughes et al. 2022). In addressing rehabilitation, it places strong emphasis on principles related to human rights, human dignity, and human agency.

Specifically, GLM focuses on what have been termed 'prudential values' that are related to the individual's level of quality of life and well-being (Day et al. 2019). That is not to say that the GLM ignores risk factors for crime, but it recognises that offending is often a means to achieve primary goals which all humans may seek. In effect, people who commit

crimes are "people like us" (Ward and Durrant 2021, p. 8). All people seek power, resources, or other primary psychological "goods" such as respect and admiration (or other natural desires such as pleasure). While some people pursue these goals through pro-social norms, other people engage in offending behaviour (Ward 2002). Therefore, rehabilitation involves developing a 'good lives plan' to understand the person's holistic self and priorities, not just the individual's offending circumstances. The rehabilitation plan addresses barriers plus methods to achieve the goals, rather than focusing on the person's deficits.

The GLM has also been analysed in clinical and community rehabilitation settings for chronic and complex health conditions (Siegert et al. 2007). For example, rather than simply focusing on symptom removal or symptom management (e.g., reducing pain in health conditions) or short-term secondary goals (e.g., obtaining employment), a GLM approach encompasses broader life primary goals such as social connection and well-being. Critics of the GLM point to shortcomings in evidence of reducing recidivism (e.g., Looman and Abracen 2013; Zeccola et al. 2021), though more recent research suggests that when GLM theory is tightly applied and aligned with the core theoretical components, there is evidence of risk reduction (Day et al. 2019).

Applied in the context of professional misconduct, while disciplined, the nature of the rehabilitation penalty would be informed by a series of assessments. This includes a focus on the practitioner's circumstances and 'ways of living' at the time of the misconduct, incorporating both subjective judgements made by the practitioner and external (expert) judgements made by an assessor. Such information-gathering would illuminate pressing needs, obstacles, risks, and pathways/factors that contributed to the misconduct. Moreover, positive elements to be utilised would also be noted. A mapping table can be used to document and inform the capacity, means, and scope of the rehabilitation content. This includes short-term and long-term goals as part of a 'life plan' to realise the person's goals. For example, in the context of sexual misconduct, this might include a review of past and present intimate relationships, and future relationship needs and priorities. Methods and skills needed to realise these goals in an ethical way would be part of the life plan.

We were unable to find any explicit reference to GLM theory in professional misconduct rehabilitation contexts. However, tribunals clearly consider practitioners' well-being to varying degrees, but do not refer to explicit frameworks such as the GLM. For example, in the case of a medical practitioner found guilty of professional conduct relating to codeine use and unauthorised prescribing, a disciplinary tribunal described the offending as "very much a product of her addiction" and that understanding "the root of the offending" directly informed their view that "a path available for rehabilitation" was available (https://www.hpdt.org.nz/portals/0/636Med14272P.pdf, accessed on 8 February 2023). Then the tribunal delegated the details of implementation to a health committee.

### 3.2.4. Therapeutic Jurisprudence

Therapeutic jurisprudence (TJ) has attracted significant attention in recent decades, initially being advanced in mental health law contexts (Wexler 2014), it is concerned with:

> "The study of the role of law as a therapeutic agent. It is an interdisciplinary approach designed to produce scholarship that is particularly useful for law reform. (It) proposes the exploration of ways in which, consistent with principles of justice, the knowledge, theories and insights of the mental health and related disciplines can help shape the development of the law." (Wexler and Winick 1996, p. xvii)

Proponents of TJ have championed its merits (Freckelton 2008a) and much has been published regarding its impacts across contexts and jurisdictions (e.g., Brookbanks 2015; Diesfeld and Freckelton 2003; Perlin 2008). Early critiques of TJ included that it was poorly defined and lacked a coherent theory (Slobogin 1995).

In brief, TJ encourages the use of therapeutic approaches, for example to sentencing, which in turn supports the principle of rehabilitation, especially in regard to psychological and social well-being (Jones 2021).

Subsequently, scholars have attempted to articulate its core components (e.g., harnessing empathy, respect, and non-paternalism in sentencing, Goldberg 2011). Arising from this, recent writings describe TJ as a testable theory (Vols 2019) prompting research efforts to standardise the measurement of the core components (Kawalek 2020). These include providing therapeutic support by decision-makers to show compassion for the legal subject's life circumstances. Another example includes therapeutic engagement through sincere, honest dialogue with the person. Moreover, the philosophy promotes constructive processes through, for example, crafting legal outcomes that build upon individuals' strengths. Developing validated tools to measure TJ is an important step in evaluating intended effects. TJ's potential also has been considered beyond the contexts of mental health, disability, and criminal law (Marinos and Whittingham 2019; Perlin 2019), to the discipline of professionals (Freckelton 2008b; Jones 2021).

In applying TJ practices to professional misconduct, disciplinary outcomes arise, but all steps in the disciplinary process (e.g., rules and procedures) would require a therapeutic lens. This includes recognition that not only are mental health or psychological factors linked with misconduct, but the disciplinary process itself also has health repercussions. Thus, it may be less adversarial in its style. Further, there may be explicit consideration of the long-term effect of discipline on the practitioner's career as well as that on their family. The penalty would not just apply to the usual approach of getting the practitioner back to safe and competent practice (e.g., the conventional considerations of protecting the public through requiring supervision or limits on practice), but would extend to what other therapeutic endeavours support the practitioner. In short, any penalty imposed, and the tribunal processes leading to this, would require consideration of their potential anti-therapeutic effects.

Possible elements of TJ already are applied by disciplinary bodies, but not in a conscious, explicit, or systematic manner, and certainly not in regard to incorporating TJ into tribunal *processes* (e.g., hearings; "different lawyering" Wexler 2012, p. 780) and *outputs* (e.g., the tone and content of written decisions). A need for a pro-therapeutic orientation, at least in regard to lawyer discipline, has been argued (Jones 2021). Given TJ's broad definition and wide application across legal contexts, it may offer some foundations for a model of professional rehabilitation. TJ's interdisciplinary foundations, and promotion of pro-therapeutic proceedings and practices, may contribute to a model of rehabilitation for disciplined professionals.

Psychological Jurisprudence

Others have described TJ as one of many forms of 'psychological jurisprudence' (PJ) practice frameworks (Arrigo 2004; Sellers and Arrigo 2022) for applying the law in a more humanistic manner. PJ draws upon diverse streams of critical studies, including literary criticism, feminist theory, psychoanalysis, political economy analysis, and postmodernism (Arrigo 2004). It proposes how the law-psychology field can promote the goals of justice and humanism in legal theory, research, practice, and policy (Arrigo 2004).

This, too, can inform future rehabilitative strategies for disciplined professionals in the civil context. For example, humanistic strategies for corrections and offender treatment have been advanced (Polizzi et al. 2014). Similarly, critiques have extended to forensic psychology in the context of mental health law and proceedings (Polizzi and Draper 2015). This body of research refers, and contributes, to critical legal scholarship regarding the origins of law, transgression, and deviance (Crewe 2013). For example, there has been a call for the use of narrative approaches based on psycho-social theory to better understand crime and punishment, rather than to predict and control it (Presser and Sandberg 2015). These efforts to constructively problematise the law are broadly aligned with the goals of those who developed the concept of psychological jurisprudence (PJ).

Those embarking on development of rehabilitation theory are reminded of PJ's call for critical examination of fundamental concepts and institutions (Arrigo 2013; Arrigo 2015; Sellers and Arrigo 2022). Creation of a theory, if based on the above and other frameworks,

should interrogate their fundamental underpinnings. The questions might include: who is defining "good lives"; what are the underpinning assumptions of a pro-therapeutic legal regime (and what is sacrificed under such a regime); how does theory meaningfully address the roles and powers of those who determine the penalty, monitor the rehabilitation, and deliver it?

## 4. Why Is a Professional Misconduct Rehabilitation Theory Needed, and What Might Be Important in That Theory?

We now turn to the purpose of theory, and the benefits of producing a professional rehabilitation theory. Theory-building and theory-testing are often neglected. They lag behind rehabilitation practice in many settings, prompting calls for more rehabilitation theories (Wade 2015) to expand the "science of rehabilitation" (Siegert et al. 2007). A clear theoretical framework is vital in the rehabilitation context:

Evidence is always viewed through the prism of theory, whether the theory is private, implicit, and unarticulated or public, explicit, and clearly articulated. An articulated theoretical framework is preferred because the explanatory and predictive features are essential to the judicious design and selection of assessments and interventions (Vaz et al. 2017, p. 121).

The underlying principles, values, and core assumptions of professional misconduct rehabilitation practices have yet to be explicitly articulated. In turn, this stymies the ability for these to be evaluated and constructively critiqued. There are potential repercussions arising from this.

First, professional discipline rehabilitation may be occurring in a theoretical vacuum, at worst. The void may result in inconsistent decisions by disciplinary bodies and decision-making that currently rely on borrowing assumptions and values from theory developed elsewhere. This borrowing may be intentional, involuntary, or random. For example, measures and descriptors of rehabilitation within some decisions appear to adopt a deficit-correcting lens; it focuses on completing further training or activities such as supervision (e.g., Mescall et al. 2017). There are also assumptions about the duration needed to achieve rehabilitation, for example when conditions are placed on practice for a specified duration. This 'deficit' and 'dose-response' approach has elements of the RNR and health models ('rehabilitation intensiveness'). However, in other cases, rehabilitation orders may be occurring with reference to some other 'private theory', which does not promote consistency or transparency.

Second, there are already mismatches in expectations amongst stakeholders which affect the acceptance and uptake of rehabilitation. A practitioner breaching sexual boundaries may believe that they are better rehabilitated by a short ethics course, rather than psychological 'treatment'. This mismatch may be exacerbated when different theories of rehabilitation are applied across decisions. However, a *theory-driven* penalty may be better understood, accepted, and effective. For example, a PCR approach would defend placing considerable weight on the practitioner's preferred means of being rehabilitated. In contrast, an RNR approach would be informed by a risk assessment conducted by experts, which may or may not align with the consumer's or practitioner's wishes. Whereas an RJ approach may prioritise outcomes valued by the victim. Yet a TJ approach may draw upon the fields of mental health and psychology to achieve pro-therapeutic processes and outcomes for the practitioner and the those affected by the misconduct. PJ may encourage examination of the institutions and structures that fostered the misconduct. As another manifestation of mismatch, a sexual boundary violation managed in the criminal context may come with different stakeholder expectations than the same conduct in a professional context. These examples highlight that different theories have some overlapping but also unique expectations and values in the rehabilitation process (Day et al. 2019). Clarification of these expectations and values helps explain decisions of disciplinary bodies.

Third, we suggest the absence of explicit theory hinders empirical research on professional misconduct rehabilitation. Rehabilitation orders are the least researched area of

professional discipline (Kiel 2017) and there are existing concerns that some penalties that place conditions on a practitioner's practice are not fit for the purpose (Paterson 2017). Developing a rehabilitation theory is the foundation for generating rigorous hypotheses, as seen in health and criminal rehabilitation contexts. Without a theory, research on this element of discipline is likely to remain stagnant, and fail to address the identified problem that professional regulation research currently includes "weak data" (Browne et al. 2021, p. 1).

*Requirements of a Professional Misconduct Rehabilitation Theory*

There are many definitions of a theory. At a literal level, it contains a set of interrelated principles, constructs, or ideas which can be used to explain or account for a phenomenon. A professional misconduct rehabilitation theory should also have logical consistency. If there are competing theories, a good theory focuses on establishing what is real (and based on real life situations) and testable, should make progress, displace myths or assumptions, and potentially inspire new ways of thinking about current rehabilitation penalty practices. Robust rehabilitation theory also needs to include an organising framework that is readily understandable to those implementing rehabilitation.

Also, good theory favours generality over specificity for particular rehabilitation issues and is open to refinement (Dunn and Elliott 2008). This means that it must also be articulated in a sufficiently high order as to be able to consider nuances between regulated professions, even for the same act of misconduct. For example, a consensual sexual relationship between a lawyer and their recent former client may have quite different misconduct considerations than the same type of conduct between a psychiatrist and their recent former patient or a teacher and their recent former pupil. Codes of ethics, duty of care, harm caused, and victim vulnerability clearly differ, but the higher principles in considering or approaching rehabilitation can be the same. We argue that such nuances equally apply when, for example, a particular rehabilitation theory is applied across different criminal contexts, or a particular theory is favoured over another in various health contexts.

We refer to Siegert et al. (2007) for description of other essential theory components. Adapting these to disciplinary rehabilitation, these include: (a) describing a set of principles and assumptions specifying the values underlying rehabilitation practices, processes, and aims for which those ordering rehabilitation should aim for; (b) factors which can explain the professional misconduct and how the rehabilitation can influence these; and (c) the implications for these components in deciding the most suitable style/format/content of rehabilitation orders.

We anticipate that a professional rehabilitation theory would likely draw on interdisciplinary concepts from psychology, law, and socio-cultural fields. For example, it is likely to incorporate theories regarding how people can change, including psychological (e.g., developing insight) and pedagogical (e.g., is the deficit 'learnable') constructs. However, other components, and their prioritisations, may be novel. As noted above, the relevant concepts values and perspectives should aim to be applicable across a range of practitioner professions despite some means of misconduct likely to be profession-specific (doctors and pharmacists having prescribing-related misconduct; lawyers failing to disclose relevant facts or conflicts of interests).

We also propose that mixed methods research (MMR) is well-placed for exploring a rehabilitation theory. Quantitative research can explore current rehabilitation practices to reveal patterns of when, why, and what rehabilitation penalties are ordered. Qualitative research can explore the assumptions made by those ordering rehabilitation, the assumptions and experiences of those who provide the service, and the experience of those who undergo rehabilitation. These stakeholders' views can reveal their rehabilitation priorities, values, preferred processes, and reasons for these. MMR is especially helpful in accessing and then utilising perspectives to understand complex social issues and phenomena (Tashakkori and Creswell 2007; Anderson 2016). In short, as has occurred in other rehabilitation contexts, efforts to generate professional misconduct rehabilitation theory will necessarily involve programmes of research conducted by many, and evolve over time.

## 5. Conclusions

In conclusion, we agree with the insight of (Elkin et al. 2012) regarding the discipline of health practitioners:

"In sum, tribunals appear inclined to look behind the labels used in the charges brought before them to evaluate the rehabilitative potential of the doctor concerned. Research into professional misconduct must therefore do the same if it is to reach the aspects of clinicians' behaviour that drive regulators' deepest public-safety concerns and shape their calculus around sanctions." (p. 1032)

In this paper we suggest that "look(ing) behind the labels" must also extend to development of a theory about misconduct rehabilitation generally. Theory aids the precision, consistency, transparency, and hopefully the success of rehabilitation for the benefit of all, including those who are served by our regulated practitioners. We welcome wider debate and dialogue on this issue.

**Author Contributions:** Conceptualisation, L.J.S., K.D. and M.R.; writing—original draft preparation, L.J.S.; writing—review and editing, L.J.S., K.D. and M.R. All authors have read and agreed to the published version of the manuscript.

**Funding:** The ideas expressed in this paper were developed as part of a Marsden Fund, Royal Society of New Zealand application 22-AUT-015 subsequently awarded in November 2022.

**Institutional Review Board Statement:** Not applicable.

**Informed Consent Statement:** Not applicable.

**Data Availability Statement:** Not applicable.

**Conflicts of Interest:** The authors declare no conflict of interest.

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
