# Peer review of "Practitioner Rehabilitation following Professional Misconduct: A Common Practice Now in Need of a Theory?"

_laws, 2009_

Round 1
Reviewer 1 Report
The authors offer a very promising idea but, regrettably, deliver very little that's new. They summarize leading theories of rehabilitation from the health and justice literatures as a basis to to consider theorizing rehabilitation for practitioners who engage in misconduct. I very much like this idea. That said, the bulk of the paper must be about theorizing rehabilitation for practitioners who engage in misconduct. Without this particular focus, the paper significantly lacks originality.
In the paper's first major section, the authors miss the opportunity to include some of the more humanistic (e.g., Dave Polizzi or Lois Presser or Ben Crew) and critical (Arrigo and colleagues) theoretical work in "offender" therapy, treatment, and recovery. Arrigo and colleagues have even developed the subfield of "psychological jurisprudence.This theoretical lens should be included. The collective authors ask important questions about theorizing rehabilitation. For example:
"good lives" according to whom?
"restore" based on what terms or conditions?
"therapeutic" as derived from which value system of justice?
But, more importantly and regrettably, the authors do very little to advance the conversation, regarding a theory of rehabilitation for institutional and community practitioner's of "offender" behavior and/or misconduct. Here, we're speaking not about "the kept," but instead, refer to "the keepers or the kept" and the "watchers" of the watched." A useful starting point might be:
Arrigo and Milovanovic (2009). Revolution in Penology: Rethinking the Society of Captives.
Additional theoretical (as well as clinical, and diagnostic) work is found at:
Arrigo (2013). "Managing risk and marginalizing identities: on the society of captives thesis and the harm of social dis-ease." International Journal of Offender Therapy and Comparative Criminology, 57(6): 672-693.
Arrigo, (2015). "Responding to crime: Psychological jurisprudence, normative philosophy, and trans-desistance theory." Criminal Justice and Behavior, 42(1): 7-18.
Sellers and Arrigo. The narrative framework of psychological jurisprudence: Virtue ethics as criminal justice practice. Aggression and Violent Behavior, 63, 2022 101671.
Bottom line: the authors of have a great idea but originality is lacking. I would love to read a revision if the authors are willing to undertake the mid-level revision work.
Author Response
Please see File attached

Reviewer 2 Report
This is an interesting article that provides an overview of several rehabilitative (and related) theories in arguing that a single theory is needed for the rehabilitation of professionals. Although the paper does not provide the theory, the authors provide what they believe are the minimum requirements of such a theory. This is an important contribution to the literature that can be used in testing theories proposed in the future. However, it would be beneficial to go into greater detail on the requirements and talk specifically about different types of professionals.
Overall, it is an interesting and well written paper. However, there are several areas where the paper could be further improved. I detail some suggested edits by section below.
Section 1 (Introduction)
The second paragraph of the introduction (Line 35) states “In New Zealand”. I encourage the author to delete this as the paper is not about any specific jurisdiction but mentioning a single jurisdiction that early in the paper signifies that the paper is specific to New Zealand and will likely dissuade many readers from other jurisdictions from continuing with the article. New Zealand is also mentioned on lines 151 and 160-161. Although this is fine, I encourage the author to also make reference to other jurisdictions.
Section 2
Although not strictly defined, there is an inconsistency between saying that rehabilitation “involves restoring someone to a previous, relatively better state” (lines 24-25) and saying that “rehabilitation will impact the speed, degree or type of ‘recovery’”. It is clear what the author is intending to say, but the terminology requires refining. Perhaps change “rehabilitation” to “rehabilitative theory" in line 57?
Section 3
On lines 130-131 the term ‘get over’ can be interpreted in multiple ways. For example, it could mean that the harm to their career cannot be repaired or they cannot personally recover from the emotional impact of a past action. The former seems the most logical interpretation of what the authors intend but I suggest editing the line.
The inclusion of restorative justice as a related theoretical model makes sense. However, RJ is not itself truly a rehabilitative model, at least not from the perspective of the offender. The model focuses primarily on victims and, to the extent it focuses of the offender, its assistance in their integration into society is more about the acceptance of victims/society than the self recovery of the offender. The focus on society and repairing relationship can, of course, be quite beneficial to the offender and in a professional context it may facilitate the offender’s reintroduction into a professional role (which is why it is relevant here) but the core goals differ from traditional rehabilitative theories. I suggest making this distinction clear in the text.
Line 219 correctly states that one critique of GLM is lack of “evidence of reducing recidivism.” However, the article citied (Looman & Abracen 2013) specifically focuses on sexual offenders. As sexual offences are less likely to be a significant portion of professional rehabilitation efforts, it is important to note the limited scope of the cited literature before considering the model’s applicability to professional contexts.
Section 4 (Significance)
The second issue addressed is that of a mismatch “in expectations amongst stakeholders” (line 289). A discussion about the difference in stakeholders between a professional context and stakeholders in the medical and criminal contexts of rehabilitation would be useful.
The overall discussion would benefit from either focusing on a particular class of professionals or applying theories to multiple classes. Not only are the basis for disqualification or penalty different for different types of professionals, but the overall risk factors are also different. For example, page 8 uses the example of a “practitioner breaching sexual boundaries.” This would of course be problematic for professionals like accountants but substantially more so for medical professionals whose clients may be in more vulnerable positions. Perhaps the same theory can be applied to all professions but the degree of necessary rehabilitative treatment varies. However, recognising the distinction is important.
4.1. In the context to fiduciary relationships (covering a wide variety of professionals), different jurisdictions can set a variety of standards but these can be summarised under two broad categories: A duty of care (ensure the client receives careful and competent advice/treatment) and a duty of loyalty (act in the clients best interests). Breach of these duties will often result in the loss of a fiduciary’s ability to continue in their professional capacity but it is quite different from a health issue or sexual misconduct. As such, a critique of rehabilitative theory for professionals should address these types breaches of professional standards.
Section 5 (Conclusion)
The conclusion’s quote specifically focuses on the rehabilitation of a doctor (presumably meaning physician). The entire paper was about professionals generally but this appears to limit the scope to medical professionals.
The authors provide an overview of many theories, including their strengths and weaknesses. Although not strictly necessary, a table detailing the benefits of models and the characteristics necessary for a universal rehabilitative theory would be beneficial. This could serve as an important reference against which future researchers may test proposed theories.
Minor grammar and citation issues to address
There is an extra period in the middle of line 70
There is an extra comma at the end of line 142
There is a missing citation at the end of line 124
Author Response
Please see attached reply file

Round 2
Reviewer 1 Report
The authors have now addressed my concerns. Thank you.